# Methyl Gallate Alleviates Acute Ulcerative Colitis by Modulating Gut Microbiota and Inhibiting TLR4/NF-κB Pathway

**DOI:** 10.3390/ijms232214024

**Published:** 2022-11-14

**Authors:** Ping Zhou, Jia Lai, Yueyue Li, Junzhu Deng, Chunling Zhao, Qianqian Huang, Fei Yang, Shuo Yang, Yuesong Wu, Xiaoqin Tang, Feihong Huang, Long Wang, Xinwu Huang, Wenjun Zou, Jianming Wu

**Affiliations:** 1School of Pharmacy, Chengdu University of Traditional Chinese Medicine, Chengdu 611137, China; 2School of Basic Medical Sciences, Southwest Medical University, Luzhou 646000, China; 3School of Pharmacy, Southwest Medical University, Luzhou 646000, China; 4Luzhou Key Laboratory of Activity Screening and Druggability Evaluation for Chinese Materia Medica, Southwest Medical University, Luzhou 646000, China; 5Education Ministry Key Laboratory of Medical Electrophysiology, Southwest Medical University, Luzhou 646000, China

**Keywords:** ulcerative colitis, methyl gallate, gut microbiota, gut immunity, TLR4/NF-κB

## Abstract

Ulcerative colitis (UC) is a complex immune-mediated inflammatory disease. In recent years, the incidence of UC has increased rapidly, however, its exact etiology and mechanism are still unclear. Based on the definite anti-inflammatory and antibacterial activities of *Sanguisorba officinalis* L., we studied its monomer, methyl gallate (MG). In this study, we employed flow cytometry and detected nitric oxide production, finding MG regulated macrophage polarization and inhibited the expression of proinflammatory cytokines in vitro. MG also exhibited anti-inflammatory activity accompanying with ameliorating body weight loss, improving colon length and histological damage in dextran sulfate sodium-induced UC mice. Meanwhile, transcription sequencing and 16S rRNA sequencing analyzed the key signaling pathways and changes in the gut microbiota of MG for UC treatment, proving that MG could alleviate inflammation by regulating the TLR4/NF-κB pathway in vivo and in vitro. Additionally, MG altered the diversity and composition of the gut microbiota and changed the abundance of metabolic products. In conclusion, our results are the first to demonstrate that MG has obvious therapeutic effects against acute UC, which is related to macrophage polarization, improved intestinal flora dysbiosis and inhibition of TLR4/NF-κB signaling pathway, and MG may be a promising therapeutic agent for UC treatment.

## 1. Introduction

Ulcerative colitis (UC), a chronic and immune-mediated inflammatory disease, starts in the rectum and extends proximally in a continuous manner through the colon [1,2]. The characteristics of UC are a long course of disease, suffering from repeated attacks and not easy to heal, and UC has symptoms of abdominal pain, bloody diarrhea, weight loss [3]. According to epidemiology studies, since the 21st century, UC has shown a high incidence in developed countries and a dramatic increase in developing countries, and it has evolved into a global disease, posing a major challenge to health care systems worldwide [4,5,6].

As a subclass of inflammatory bowel disease (IBD), UC is characterized by persistent inflammation in the intestinal mucosa or submucosa. These inflammatory reactions are due to the loss of immune tolerance. Macrophages in the mucosa of the gastrointestinal tract polarize to M1 phenotype in UC patients, secreting a large number of proinflammatory cytokines to enhance cytotoxicity and phagocytosis [7], then resulting in direct damage to epithelial cells and the intestinal barrier. Meanwhile, the pathogenesis of UC involves immune-inflammatory pathways related to various intestinal components, including Toll-like receptors (TLRs) and inflammatory factors such as TNFα, IL-1 and IL-6 [1,8]. Therefore, once the inflammatory treatment of UC was inadequate, it would lead to persistent intestinal damage and increase the risk of hospitalization, surgery and colorectal cancer [9]. Additionally, studies have shown that the destruction of the intestinal mucosal barrier and the abnormal activation of immune system also involve the imbalance of intestinal flora, and a great number of differential metabolites produced in UC patients is closely related to the imbalance of intestinal flora [10,11]. Therefore, maintaining the homeostasis of intestinal flora is also crucial for the treatment of UC.

Treatment for UC is thought to primarily focus on reducing inflammatory responses, increasing intestinal mucosal healing, and strengthening the intestinal mucosal barrier during this stage [12]. Furthermore, in terms of pathophysiology, the treatment of UC is mainly manifested as follows: (1) the colon epithelium promotes the interaction between the host and microorganism, thus controlling mucosal immunity, coordinating nutrient circulation and forming a mucus barrier; (2) homeostatic balance among host mucosal immunity, function and composition of intestinal microbiota; (3) complex innate and adaptive in immunomodulatory mechanisms in the gastrointestinal tract, including immune cells such as macrophages and neutrophils, and non-immune cells such as intestinal epithelial cells (IEC) and mesenchymal cells, intestinal epithelial cells possess a wide range of pattern recognition receptors, including Toll-like receptors (TLR) and NOD-like receptors, which are the key components of the innate immune system. Most drugs used to treat UC could interfere with metabolic and immune responses and still have limitations, including severe side effects, tumorigenesis and high costs [13,14]. Therefore, there is a need to develop more effective, less toxic and cheaper drugs to benefit patients suffering from UC.

*Sanguisorba officinalis* L. (SOL) is a Rosaceae plant with hemostatic, anti-inflammatory and astringent properties [15]. *Sanguisorba officinalis* L., traditional Chinese medicine, has been used in clinic for thousands of years. Many compound preparations, including SOL, have shown good clinical therapeutic effects on UC [16,17]. However, because the pathogenesis of UC is complex, the mechanism of SOL treating UC is still unclear, and the components of traditional Chinese medicine are complex and difficult to predict, we chose to extract monomers from SOL for activity screening, in which methyl gallate (MG) showed high anti-UC activity. MG is a polyphenolic compound extracted from SOL with the molecular formula C_8_H_8_O_5_. Recent studies have shown that MG is widely distributed in medicinal and edible plants and is considered to have medicinal properties such as antioxidant [18,19], antitumor [20], anti-inflammatory [21], and antibacterial properties [22,23]. In addition, it is inexpensive and easily available. However, the role of MG in the treatment of UC has been rarely studied, and the specific molecular mechanisms are still unclear.

In this study, we constructed disease models in vivo and in vitro to clarify the efficacy of MG on UC and the molecular mechanisms of the phenotypic, protein, mRNA, and gene levels. Furthermore, we applied high-throughput genomic analysis techniques, in which transcriptome and microbial diversity might provide insight into transcriptional regulation of genes and changes in gut microbiota, respectively. Our findings may lead to providing a potential therapeutic strategy with low cost, low toxicity and high efficacy for a growing number of UC patients.

## 2. Results

### 2.1. MG Alleviated Inflammatory Response in the Lipopolysaccharide (LPS)-Induced UC Model In Vitro

After screening for drug activity, given that methyl gallate displayed an effective improvement in nitric oxide releasing of inflammatory model in vitro. We conducted cytotoxicity assays with MTT and analyzed total NO production for MG under time- and concentration-gradient conditions. The results are shown in Figure 1A–D. We limited the concentrations of MG and Sulfasalazine (SASP) in the subsequent experiments to below 80 µg/mL and 800 µg/mL, respectively, and the culture was maintained for 24 h. To better investigate the effect of MG (5, 10 and 20 μg/mL) on the total level of NO production in LPS-induced RAW264.7 cells, we evaluated the changes in NO production after 12 and 24 h of MG (5, 10 and 20 μg/mL) and LPS (1 μg/mL) intervention treatments. As shown in Figure 1E,F, the release of total NO production in LPS-induced RAW264.7 cells were significantly inhibited after 10 and 20 μg/mL MG administrations for 24 h (*p* < 0.05). The results suggested that MG (5, 10 and 20 μg/mL) inhibited LPS-induced NO generation in model RAW264.7 cells. Furthermore, in view of the subsequent concentration based on the above experimental results, we used a 24 h intervention with 1 µg/mL LPS and 5, 10, and 20 µg/mL MG as the experimental concentration protocol.

To verify the role of MG, we assessed macrophage polarization by detecting the positive expression of M1 and M2 macrophage markers by flow cytometry. As demonstrated in Figure 2, under LPS stimulation, LPS conspicuously promoted the expression of the M1-like macrophage marker CD16/32 in labeled macrophages (*p* < 0.001) and slightly decreased the expression of the M2-like macrophage marker CD206 in macrophages under LPS stimulation. In contrast, high doses of MG (20 μg/mL) caused a significant reduction in M1-like macrophage cells in macrophages, while the proportion of M2-like macrophage cells increased significantly (*p* < 0.05). To further determine whether the anti-inflammatory effects of MG are mediated by macrophage regulation, we detected the inflammatory cytokines that are involved in phenotypes of M1 and M2 macrophages. Furthermore, MG treatment dramatically decreased the expressions of IL-1β, IL-6, and TNF-α in LPS-induced macrophages (Figure 2C). Notably, MG treatment significantly promoted the expressions of Arg-1, IL-4, and IL-10, which acted as M2-mediated specific markers in LPS-stimulated RAW264.7 macrophages. These results suggest that MG could effectively promote the polarization of M1-like macrophages to M2-like macrophages and inhibit the release of inflammatory cytokines.

### 2.2. MG Alleviated the Inflammatory Response of Dextran Sulphate Sodium (DSS)-Induced UC Model Mice

Given that MG had an anti-inflammatory effect in vitro, we thereby evaluated whether it had therapeutic effects on mice. In our study, mice in model group and treatment groups were administrated with drinking water containing 2.5% (*w*/*v*) dextran sulphate sodium (DSS) for 8 days to induce acute colitis. During the experiment, mice in control group remained in a normal state, while those mice in model group presented loss of appetite, weight reduction and persistent rectal bleeding (Appendix A), which were typical features in DSS-induced ulcerative colitis model. To assess the treatment effect of MG on UC in vivo, body weights, DAI scores, and changes in colon lengths of mice in each group were analyzed. As shown in Figure 3B–E, DSS-induced UC mice exhibited significant weight loss, increased DAI scores and colon shortening, which was clearly shown in Appendix A. However, those that were administered MG (50, 100 and 200 mg/kg) and SASP (500 mg/kg) were significantly resistant to DSS-induced UC by all of these measures.

Consistently, as shown in Figure 3A,B, no ulcer was identified in the colon tissue of the normal group, then, the glands were in regular alignment, and the mucosa was complete. However, DSS led the colonic mucosa to defective, glandular structure to separate, and a considerable number of neutrophils to be infiltrated. In addition, it was demonstrated that there are apparent inflammatory reactions, including granulation tissue hyperplasia and fibrosis, and obvious lesions that were deeply localized in the muscle layer and the serosal layer. In conclusion, compared with the model group, oral administration of MG (50, 100 and 200 mg/kg) significantly alleviated the histologic damage of model mice in a concentration-dependent manner.

Furthermore, we observed pathological changes in the liver (Appendix A) and kidney (Appendix A) to assess the effects of DSS and MG on the histological structure of mice. The results are similar to a recent report [24], with the exception of the normal group exhibiting normal liver histology, there were various degrees of damage in DSS-induced model and treatment groups, among which the pathological changes of inflammatory infiltration and necrosis were most significant in model group. Additionally, kidney tissue lesions were observed in DSS-induced model and treatment mice, such as glomerular inflammation and congestion, as shown in Appendix A, medium-high dose of MG-treated and SASP-treated groups had kidney histological structures which were similar to those of the normal mice. Specifically, the glomerular morphologies were more regular, the cystic cavities were visible, and the pathological degrees of inflammatory infiltration and fibrose were lower in the renal interstitium. By analyzing the pathological section of each tissue organ, it was found that MG could alleviate the damage to the liver, kidney, and colon histological structures caused by DSS.

### 2.3. Bioinformatics Analysis Indicated the Underlying Biological Function of MG in DSS-Induced Ulcerative Colitis

Transcriptome analysis was performed to determine the underlying molecules regulated by MG to explore the potential mechanism of MG against UC. As shown in Figure 4A, when evaluating differential gene expression and displaying the volcanoes of differential genes, the closer the points toward the discrete edge are, the more important the genes have different expression. We could obtain co-expressed and differentially expressed genes or transcripts among groups, and the value represents the number of common and differentially expressed genes or transcripts among different groups. Additionally, quantifying the proportion of expressed genes makes it more intuitive to compare the differences in expression among groups. Based on the results of the statistical analysis of differentially expressed genes, negative binomial distribution software such as DESeq2, DEGseq and edgeR were used, and *p* < 0.05 and |log2FC| ≥ 1 were used for the screening conditions, we compared the normal group with the DSS-induced model group in terms of the expression of genes (Figure 4B), the differentially expressed genes are roughly symmetrically distributed on the volcano diagram, and the closer the point is scattered to the edge, the more significant the difference in genes expressed. A total of 1159 differentially expressed genes were screened, of which 814 genes were significantly upregulated and 315 genes were significantly downregulated. Then, as for gene expression profile of colons in MG treatment groups, 522 genes were considerably modified compared to the DSS model mice, 100 of which were highly upregulated and 422 of which were significantly downregulated. We screened out the genes with significant expression from them for subsequent research.

After the differentially expressed genes were screened out, we further carried out enrichment analysis of GO and KEGG pathways of different genes to elucidate the functions of differentially expressed genes. In this study, we investigated the pathways of GO enrichment among groups using multi-point enrichment analysis, and the results are displayed in Figure 4C. The differentially expressed genes were significantly enriched in biological processes, including granulocyte migration, regulation of acute inflammatory response, and regulation of neutrophil migration in both the model and treatment groups. Similarly, we performed a multi-gene set enrichment analysis of the KEGG pathway, which showed significant enrichment of the NF-κB signaling pathway, intestinal immune network for IgA production and cytokine‒cytokine receptor interaction, as detailed in Figure 4D. The above enrichment results suggest that treatments with DSS and MG affect the expression of pathways linked to the regulation of inflammatory reactions, NF-κB signaling and the production of IL-4, which might be connected to the potential molecular mechanism of MG treatment effect on UC.

### 2.4. The Mechanisms of MG Inhibiting Gut Immunity, Macrophage Infiltration and Inflammatory Cytokines

There is growing evidence that proinflammatory cytokines play a crucial role in DSS-induced UC [7,25,26,27], including TNF-α, IL-1β, and IL-6, which is consistent with the results enriched by our transcriptome sequencing. qPCR was used to evaluate the relative mRNA expression levels of infiltration and inflammatory cytokines. The results were consistent with the transcriptome sequencing in mice with DSS-induced colitis. In DSS-induced model group, the relative mRNA expression levels of pro-inflammatory factors, such as TNF-α, IL-6 and IL-1β, were significantly increased, however, the expression levels of pro-inflammatory cytokines were significantly decreased after MG intervention. In addition, MG intervention significantly upregulated the relative mRNA expression levels of anti-inflammatory genes, such as IL-4 and IL-10, suggesting that MG may alleviate UC by inhibiting the expression of pro-inflammatory genes and increasing the expression of anti-inflammatory genes (Figure 4E).

Thus far, the results demonstrated that MG was resistant to the colitis-associated inflammatory response. The regulatory proteins were measured using Western blotting to characterize the mechanism of MG in mice with DSS-induced colitis. NF-κB is a key regulator of innate immunity and tissue integrity, and nuclear translocation of NF-κB is substantially active in colitis model mice and UC patients [28,29,30]. We also observed significant activation of the NF-κB signaling pathway in DSS-induced colitis mice. Figure 5 depicts the results, compared with the normal group, DSS significantly upregulated the expression levels of IL-6, IL-1β, TNF-α, p-IκBα, p-IKKα/β, p-NF-κB, MyD88, and TLR4 in the colon tissues of model mice (*p* < 0.05). However, the colon tissue of MG (200 mg/kg) treatment, similar to SASP, significantly reduced the expression levels of p-IκBα, p-IKKα/β, p-NF-κB, MyD88, and TLR4 compared with the model group (*p* < 0.05). Thus, it is obvious that MG inhibits the production of proinflammatory cytokines by inhibiting the TLR4/NF-κB pathway.

Macrophages are primary regulators of inflammatory responses and play a central role in inflammation-associated disorders, including UC [31,32]. To determine whether MG has regulating effects on macrophages in the same mechanism as treatment groups with DSS-induced colitis in vivo, we extracted cellular proteins and characterized critical targets for validation in vitro. Western blot analyses confirmed that MG treatment in vitro reduced the expression of the proinflammatory cytokines TNF-α, IL-1β, and IL-6 (Figure 6A). Furthermore, the levels of the target proteins were measured to study the expression of critical proteins in TLR4/NF-κB signaling pathway in MG-treated macrophages. As shown in Figure 6B, the relative protein expression levels of p-IκBα, p-IKKα/β, p-NF-κB, MyD88, and TLR4 in LPS-induced model group were significantly higher than those in control group, while pretreatment with high concentrations of MG significantly reduced the phosphorylation levels of NF-κB, IκBα, and IKKα/β as well as the expression levels of MyD88 and TLR4. Next, we used TAK242, a TLR4 inhibitor, for reverse validation (Appendix A), the experiments were conducted in five groups in vitro, namely the control group, LPS-induced model group, MG-treated group (20 μg/mL), inhibitor-treated group, drug and inhibitor co-intervention group, the expression of TLR4, p-IκBα and p-NF-κB in MG-treated group and inhibitor-treated group have no significant difference, but they are lower than those in model group, indicating that MG has the same effect of anti-inflammatory pathway expression with TLR4-inhibitor, and the expression of inflammatory signal pathway in co-intervention group is similar with that in MG-treated group, indicating that TAK242 could competitively inhibit TLR4 target with MG, which reflects that MG exerts its effect through TLR4, thereby inhibiting the expression of downstream inflammatory signaling pathway and inflammatory factors. The above data suggested that MG might inhibit activation of the TLR4/NF-κB pathway in vitro, which corresponds with that in vivo.

### 2.5. MG Rectified Gut Microbiota Imbalance in Ulcerative Colitis

Since the imbalance of gut microflora is thought to be a crucial factor in DSS-induced colitis [33,34,35], and the immunomodulatory effect of MG has been established, we following asked whether MG could influence gut microbiota in DSS-induced colitis. To address this query, we analyzed the feces obtained from three groups of mice (normal, DSS, and high-dose MG treatment groups) by 16S rRNA sequencing. The gut microbial diversity was shown to be decreased in DSS-induced colitis group, which was partially restored upon MG therapy (Figure 7). The α-diversity indexes evaluating gut microbial community richness and community diversity, including Chao index and Shannon index, decreased significantly in DSS-induced mice, whereas treatment with MG effectively restored the richness and community diversity of gut microbiota (Figure 7A). The principal coordinate analysis (PCoA) based on Bray–Curtis distance is used to visually analyze the difference in composition of gut microflora. Compared with the normal group, the distribution of PCoA along PC2 in DSS model group has obvious changes, and the difference in composition of gut microflora is obvious, while MG could reduce the difference, revealing that MG could increase the diversity and richness of intestinal microbiota and improve the difference of microbiota structure in mice with UC (Figure 7B).

The relative abundances of bacteria were severely disturbed in DSS-induced mice, among which the relative abundances of the phyla Bacteroidetes decreased, while those of the phyla Verrucomicrobiota, Proteobacteria, and Actinobacteria increased (Figure 7C). However, the abundances of these main phyla did not present significant differences among the groups. The abundances of the phyla Patescibacteria, Campilobacterota, and Cyanobacteria had distinct differences between the normal group and model group according to our statistical analysis for the average of sum in the top 20 species. After MG treatment, the quantity of Cyanobacteria dramatically decreased (Figure 7D). It is worth noting that the phyla Cyanobacteria reacted to both DSS inducement and MG treatment, indicating that Cyanobacteria might be associated with MG treatment of UC.

At the genus level, DSS-induced mice displayed a depletion of *Muribaculum* and *unclassified_f_Lachnospiraceae*, which was significantly recovered by MG treatment (Figure 7E,F). It is similar to recent reports that the genera *Turicibacter*, *Erysipelotrichales*, and *Staphylococcaceae* significantly increased in DSS-induced mice [5,36]. In contrast, these pathogenic bacteria remained unchanged in MG treatment group, compared with the model group, MG can significantly reduce the abundance of *Turicibacter* and *Faecalibaculu*. These results indicated that MG could dramatically change the composition of the gut microbiota, improve the ability to produce beneficial metabolites, re-establish the gut microbiota to a normal microbial community and alleviate dysbiosis of intestinal flora in DSS-induced colitis.

### 2.6. MG May Restore Gut Immunity by Altering Gut Microbiota in Ulcerative Colitis

To investigate how MG modulated intestinal immunity and gut microbiota in colitis, we performed a correlation analysis to study the relationship between the gut microbiota and immune-inflammatory factors using Spearman’s rank correlation method. In DSS-induced colitis, it was discovered that there was an increase in harmful bacteria, such as *Turicibacter* and *Staphylococcus*, and they were positively correlated with the expression of proinflammatory cytokines and key genes of the immune-inflammatory pathways (Figure 8). In contrast, there were decreases in beneficial bacteria, such as *Muribaculum, Lactobacillus*, *unclassified_f_Lachnospiraceae*, and *Candidatus_Saccharimonas*, which were negatively correlated with the expression of proinflammatory cytokines and key genes of the immunomodulatory pathways. These results suggested that MG could regulate immune homeostasis of intestinal tracts through gut microbiota to prevent colitis in mice.

## 3. Discussion

Ulcerative colitis, a chronic and complicated inflammatory disease, is characterized by abdominal pain, diarrhea, rectal bleeding and hematochezia, which greatly affects the life quality of patients. UC has evolved into a global disease, posing major challenges to health care systems around the world [4,37,38]. Currently, clinical medications used for the treatment of UC include aminosalicylates and corticosteroids; however, some of them are expensive and have serious side effects. Therefore, there is an urgent need to develop drugs with high efficacy and low toxicity to benefit patients.

*Sanguisorba officinalis* L. (SOL) is often used to clear heat, cool blood and heal wounds [39]. Since traditional Chinese medicines are complex and difficult to predict, we chose to screen the activity of the monomers extracted from SOL, among which methyl gallate (MG) showed the potential of anti-UC. MG is widely distributed in medicinal and edible plants and is considered to have therapeutic effects, such as antioxidant, antitumor, anti-inflammatory, and antibacterial. In this study, we comprehensively investigated the therapeutic effects of MG on UC and its potential mechanisms (Figure 9). We used mice with acute UC caused by 2.5% DSS in drinking water as experimental models, then they were observed the symptoms of loss of appetite, loose stools, and loss of weight on the third day after modeling. Oral administration of MG alleviated UC by decreasing DAI scores, restoring shortened colons, and reducing tissue damage. Macrophages significantly contribute to the pathogenesis of UC [31,32]. We developed an inflammatory cell model by stimulating RAW264.7 macrophage cells with LPS and explored the appropriate modeling parameters and treatment administration conditions. The results show that MG inhibited LPS-induced NO production in supernatants of the macrophage culture, in addition, macrophage M1-like activation was associated with a large amount of NO-related production of reactive metabolites [40,41]. To assess the effect of MG on RAW264.7 cell polarization, we carried out flow cytometry and qPCR experiments. As predicted, the results show that high doses of MG (20 μg/mL) significantly reduced the percentage of M1 macrophages while increasing that of M2 macrophages. Treatment with MG downregulated the mRNA expression of M1 macrophage markers while it upregulated the levels of M2 macrophage markers, which was consistent with the results detected by flow cytometry.

To better characterize the mechanism of MG against UC, the expression and functional enrichment of differentially expressed genes were analyzed based on transcriptome sequencing. The results showed that compared with DSS-induced model group, 522 genes changed significantly in the colon tissue of MG mice, among which 100 genes were up-regulated and 422 genes were down-regulated. Through GO functional enrichment analysis, the obtained differentially expressed genes are mainly enriched in the biological processes of granulocyte migration, acute inflammatory reaction, IL-4 production and neutrophil migration. At the same time, KEGG enrichment analysis showed that differentially expressed genes were clustered in NF-κB signaling pathway, the intestinal immune network that can produce immunoglobulin A (IgA), the Fc segment receptor of immunoglobulin G (FCγR) mediated phagocytosis, complement and coagulation cascade and T cell receptor signaling pathways.

Then, 16S rRNA gene sequencing of intestinal contents revealed that the improvement of MG on UC may be due to differences in the composition of the gut microbiota changed by MG to some extent. At the level of phylum, there are only significant differences between the normal group and the model group in the abundance of Patescibacteria, Campilobacterota and Cyanobacteria, and the abundance of Cyanobacteria can be significantly reduced after MG treatment. At the same time, there are studies shown that the abundance of Cyanobacteria in diarrhea and immune diseases is significantly increased. Therefore, we think that Cyanobacteria may be related to the treatment of UC by MG. At the genus level, we found that *Lactobacillus*, *norank_f__Muribaculaceae* and *unclassified_f__Lachnospiraceae* were the main gut microorganisms in the feces of healthy mice, while the MG group enriched *unclassified_f__Lachnospiracea* and *Muribaculum*, indicating that MG improved the ability of producing beneficial metabolites and improved the imbalance of intestinal flora.

In addition, through the correlation analysis between the gut microbiota and inflammatory factors, we found that the increase in pathogenic bacteria in the model group was positively correlated with the expression of proinflammatory cytokines in UC. Compared with the model group, the abundance of Cyanobacteria, a bacterium that is positively correlated with the expression of key genes in the proinflammatory pathway, decreased significantly in the MG-treatment group, indicating that MG may regulate the dysfunctional microbiota structure and alleviate the intestinal inflammatory environment of UC mice by promoting the increase in beneficial microorganisms and the decrease in some opportunistic pathogens.

TLR4, MyD88, and their downstream signaling molecules (IκBα and NF-κB) are regarded as playing a critical role in the development of DSS-induced UC [42,43,44]. We demonstrated that MG inhibited inflammatory responses mediated by the TLR4/NF-κB signaling pathway, which was verified both in vitro and in vivo. The expression of the inflammatory markers IL-1β, TNF-α, and IL-6 increased in the colon tissue of UC mice, which was consistent with a recent report [45,46], whereas MG effectively decreased the expression of these inflammatory factors. As for validation at the cellular level, MG suppressed the production of proinflammatory cytokines and their downstream mediators, including TNF-α, iNOS, IL-1β, and IL-6. Since polarization of macrophage cells directly influences the pathogenesis of UC, it is believed that rectification of cytokine disorder may be the predominant mechanism of MG treatment, and it may be mediated by TLR4/NF-κB signaling pathway.

On the one hand, through combining transcriptome sequencing and microbial diversity, we comprehensively exploited and integrated high-throughput sequencing data to elucidate the intricate and multi-pathway pathophysiological processes in the gut, thereby revealing the regulatory mechanism of UC in a more complete manner [47,48,49]. On the other hand, through the established and exact mechanisms of action, we may investigate and predict novel targets, then find therapeutic options to improve the efficacy of clinical treatment in UC. Moreover, there was a strong correlation between the gut microbiota and inflammatory cytokines or other genes involved in immune-inflammatory pathways [50,51,52]. It appears that MG may regulate intestinal immune homeostasis through the gut microbiota in healthy as well as in disordered mice, however, the exact mechanisms deserve further investigation.

However, our research has a few drawbacks. In this study, the effect of MG on macrophage polarization was investigated in vitro, which did not fully replicate the physiological process in vivo. Although the expression of mRNA has been used to confirm the variations in proinflammatory and anti-inflammatory factors related to macrophage polarization, the types of macrophage polarization in colonic tissues have not been defined in mice. In addition, although we adopted a high-throughput sequencing approach to elucidate the physiological processes of diseases or drug treatment in organisms from multiple levels and perspectives, in this process, the expressions of many differential genes were screened, this study has not yet gone deeper and needs to be pursued further.

To summarize, oral administration of MG attenuated intestinal inflammation by inhibiting the TLR4/NF-κB signaling pathway and regulating macrophage polarization. Our findings might provide a potential therapeutic strategy with low-cost, low-toxicity therapy and definite efficacy for the growing number of UC patients.

## 4. Materials and Methods

### 4.1. Reagents and Antibodies

MG (purity > 98%) was purchased from Aladdin (Shanghai, China). DSS (molecular weight of 36 to 50 kDa) was purchased from MP Biomedicals (Santa Ana, CA, USA). Sulfasalazine was obtained from Meryer (Shanghai, China). Antibodies against myeloid differentiation factor 88 (MyD88; Lot: 67969-1-Ig), interleukin-1β (IL-1β; Lot: 66737-1-lg), interleukin-6 (IL-6; Lot: 66146-1-lg), tumor necrosis factor-alpha (TNF-α; Lot: 17590-1-AP), Toll-like receptor 4 (TLR4; Lot: 19811-1-AP), nuclear factor kappa B (NF-κB; Lot: 10745-1-AP) and GAPDH (Lot: 60004-1-IG) were purchased from Proteintech (Wuhan, China). Antibodies against β-actin (Lot: 4970), phospho-NF-κB (Lot: 3033), phospho-IκBα (Lot: 2859), p-IKKα/β (Lot: 2697), iNOS (Lot: 2982), IκBα (Lot: 4814), anti-rabbit IgG (Lot: 4414), and anti-mouse IgG (Lot: 4410) were purchased from CST (Boston, MA, USA).

### 4.2. Animals

Specific-pathogen-free (SPF) male C57BL/6J mice (6–8 weeks, 18–20 g) were obtained from Specific Pathogen-Free Biotechnology Co., Ltd. (Certificate No. 2019-0010, China). Animals were kept under SPF conditions with suitable temperature (25 ± 2 °C) and humidity (60–65%) for 12 h light/dark cycles. All operations on mice were performed in accordance with the approved guidelines of the Animal Experimentation Ethics Committee at Southwest Medical University (License NO. 20210302-011).

### 4.3. Induction of DSS Colitis in Mice and Drug Treatment

Acute colitis was induced in mice via administration of 2.5% (*w*/*v*) DSS in drinking water [53,54]. Mice received either regular drinking water (normal group) or 2.5% DSS drinking water (model group and treatment groups) for 8 days followed by regular drinking water for 6 days. Mice were randomly divided into 6 groups with 10 mice per group: the untreated normal control, model group, low-dose MG group (50 mg/kg), medium-dose MG group (100 mg/kg), high-dose MG group (200 mg/kg) and SASP-positive group (500 mg/kg) that were orally administered once daily for 6 days (Figure 1A).

### 4.4. Disease Activity Index (DAI)

To assess DAI scores, the changes of the weight and feces in mice were observed and recorded every day. The severity of colitis was evaluated by monitoring clinical manifestations daily, including body weight, stool consistency, and rectal bleeding. The DAI scores were calculated based on the well-established system shown in Appendix A, as previously confirmed [55]. The indicators of weight loss were taken for the difference between the initial and tested weights, then, the judgment standard of diarrhea was defined as the absence of fecal pellet formation and the presence of continuous fluid fecal material in the colon. The disease index of stool occult blood was assessed by a Fecal Occult Blood Diagnostic Kit. Last, DAI values were calculated as the sum scores of the weight loss value, diarrhea value, and rectal bleeding value.

### 4.5. Histological Analysis

At the end of the experiment in vivo, mice were sacrificed after deep anesthesia. Colons were exteriorized rapidly, and their lengths were measured, meanwhile, colons were preserved at −80 °C after being rinsed with ice-cold PBS. Parts of the colons fixed with 4% paraformaldehyde solution were embedded by paraffin and cut into 5 µm thick sections, then the samples were stained with hematoxylin and eosin (H&E) and observed the samples under microscope for histological evaluation.

### 4.6. Cell Culture and Treatment

The murine macrophage cell line RAW264.7 was purchased from American Type Culture Collection (ATCC, Rockville, MD, USA) and cultured with DMEM medium contained 10% FBS (FBS, CAT: SP10020500, Sperikon Life Science & Biotechnology Co., Ltd., Chengdu, China). The cells were treated with 1 μg/mL LPS in the absence or presence of MG, and all of them were incubated in a 5% CO_2_-humidified atmosphere at 37 °C.

### 4.7. Cell Viability

Cell viability was determined by using Thiazolyl blue tetrazolium bromide (MTT) assay. Cells were seeded into 96-well plates and treated with LPS. After LPS treatment for 12 h, cells were treated with or without different samples. Then, the cells were incubated with 0.5% MTT for 4 h. The medium was removed, and 100 μL of dimethyl sulfoxide solution was added. The absorbance of the reaction product in solution was measured at 570 nm using a Cytation3 microplate spectrophotometer (BioTek, Burlington, VT, USA). The percentage of living cells was calculated by the ratio of OD values.

### 4.8. Nitric Oxide (NO) Production and Quantification

The level of NO production in RAW264.7 cells was detected by a Griess reagent kit (Beyotime, Shanghai, China). cells were centrifuged at 1500 rpm for 5 min, and the supernatant fluid was transferred to new tubes, in which the supernatant was mixed with griess reagent. the absorbance of the reaction product in solution was measured at 540 nm using a Cytation3 microplate spectrophotometer (BioTek, Burlington, VT, USA)

### 4.9. Flow Cytometry Analysis

Flow cytometry was used to analyze the phenotypical changes in M1-mediated markers in RAW 264.7 macrophages. Cells were centrifuged at 1500 rpm for 5 min and washed once with ice-cold PBS, then incubated them with Alexa Fluor^®^ 647 anti-mouse CD206 (Becton Dickinson and Company, Franklin, NJ, USA), FITC-conjugated anti-mouse CD16/32 (Becton Dickinson and Company, Franklin, NJ, USA), and PE-conjugated anti-mouse F4/80 (BioLegend, San Diego, CA, USA) at room temperature in the dark for 30 min. Flow cytometry was performed using a BD FACSCanto II flow cytometer (BD Biosciences, San Jose, CA, USA). Quantitative analysis of phenotypic changes in macrophages was performed by FlowJo VX10 software (BD, Biosciences, San Jose, CA, USA).

### 4.10. RNA Preparation and qPCR Assay

TRIzol reagent (Invitrogen, Carlsbad, CA, USA) was used to extract total RNA from colon tissues and RAW264.7 cells on the basis of the instructions of the manufacturer. Single-stranded cDNA was reverse transcribed from RNA by a reverse transcription kit (Vazyme Biotech, Nanjing, China). The mRNA expression levels of proinflammatory cytokines (IL-1β, IL-6, TNFα) and anti-inflammatory cytokines (IL-4, IL-10, Arg-1) were evaluated. The primers used in this study are listed in Appendix A. The relative mRNA expression levels were normalized to the expression of β-actin. The qPCR system was as follows: 1 μg of cDNA, 10 μL of SYBR Premix Ex TaqII™ (2×), 0.2 mol/L of each primer and added ultra-pure water to 20 μL. The qPCR conditions were as follows: 95 °C for 3 min, followed by 40 cycles of denaturing at 95 °C for 30 s and 60 °C for 15 s. The qPCR data were analyzed by the 2^−∆∆Ct^ method.

### 4.11. Transcriptome Analysis

Total RNA from colon tissue in mice was extracted. One-centimeter-long samples were used for sequencing. We collected colon samples for sequencing from three randomly selected animals from the normal group, model group and MG groups, with a total of nine samples. Total RNA was extracted from the tissue using TRIzol reagent, and genomic DNA was removed using DNase I (Takara, Beijing, China). For transcription sequencing, the DNA samples were sent to Majorbio Bio-pharm Technology Co., Ltd. (Shanghai, China), under −20 °C preservation and dry ice conditions. Then, RNA quality was determined by 2100 Bioanalyser (Agilent, Palo Alto, CA, USA) and quantified using the Nanodrop2000. The RNA-seq transcriptome library was prepared following the TruSeqTM RNA Sample Preparation Kit from Illumina using 1 μg of total RNA. According to Illumina’s library construction protocol, after being quantified by TBS380, the paired-end RNA-seq sequencing library was sequenced with the Illumina HiSeq xten/NovaSeq 6000 sequencer (2 × 150 bp read length). In addition, functional enrichment analyses, including Gene Ontology (GO) and Kyoto Encyclopedia of Genes and Genomes (KEGG), were performed to identify which DEGs were significantly enriched in GO terms and metabolic pathways at Bonferroni-corrected *p* values ≤ 0.05 compared with the whole-transcriptome background.

### 4.12. 16S rRNA Microbiota Analysis

According to the present study, the E.Z.N.A.^®^ Soil DNA Kit (Omega Bio-Tek, Norcross, GA, USA) was used to extract the total genomic DNA from the colon. DNA concentrations were quantified by a Nanodrop spectrophotometer (Thermo Scientific Inc., Waltham, MA, USA), and the purity was tested on 1% agarose gel. Qualified samples were subsequently amplified the hypervariable region V3–V4 of the bacterial 16S rRNA gene with the primer pairs 338F (5′-ACTCCTACGGGAGGCAGCA-3′) and 806R (5′-GGACTACHVGGGTWTCTAAT-3′) by an ABI GeneAmp^®^ 9700 PCR thermocycler [56]. Following that, the samples were sequenced on an Illumina MiSeq PE300 platform. Operational taxonomic units (OTUs) clustering was identified at 97% sequence identity. The similarity among the microbial communities in different samples was determined by principal coordinate analysis (PCoA) based on Bray‒Curtis. The PERMANOVA test was used to assess the percentage of variation explained by the treatment along with its statistical significance using Vegan v2.5-3 package. The linear discriminant analysis (LDA) effect size (LEfSe) was performed to identify the significantly abundant taxa (phylum to genera) of bacteria among the different groups [57].

### 4.13. Western Blot Analysis

Colon tissues were homogenized in sodium dodecyl sulfate sample buffer containing proteinase and phosphatase inhibitors. RAW264.7 macrophages treated with MG were lysed with 1× RIPA (Cell Signaling Technology, Beverly, MA, USA) lysis buffer. Supernatants were collected, and protein concentrations were then measured using the BCA Protein Assay Kit (EpiZyme, Shanghai, China). The total protein samples were separated by sodium dodecyl sulfate-polyacrylamide gel electrophoresis and transferred to a PVDF membrane. The membranes were blocked in PBS-Tween 0.1% containing 10% nonfat dried milk and incubated with primary antibodies (p-NF-κB, p-IκBα, p-IKKα/β, IL-1β, IL-6, MyD88, TLR4, iNOS, TNF-α, and β-actin, all at 1:1000 dilutions, GAPDH at 1:20,000 dilutions) overnight at 4 °C. Then, the membranes were rinsed three times with PBS-Tween-20, and the secondary antibodies were added to seal for 1 h at room temperature. Signals were detected with an ECL system (Vazyme Biotech, Nanjing, China) and exposed to classic autoradiography film. After the visualization of bands for the first time, the phosphorylated proteins on the membranes were stripped by Stripping Buffer (Cwbio, Beijing, China), and the related membranes were incubated with total antibodies (NF-κB, p-IκBα) overnight at 4 °C, and the previous operation was repeated. The relative image intensities of the target proteins and phosphorylated proteins to β-actin and GAPDH positively manifested their expression.

### 4.14. Statistical Analysis

All experiments were repeated at least 3 times independently. All the data were expressed as the mean ± SEM, and one-way analysis of variance (ANOVA) was performed using SPSS v25.0 (SPSS Inc., Armonk, NY, USA) to compare the treatment means when the data were normally distributed. *p* < 0.05 was considered statistically significant. Images were processed using GraphPad Prism 8 (GraphPad Software Inc., La Jolla, CA, United States) and Adobe Photoshop CS6 (Adobe, San Jose, CA, USA).

## 5. Conclusions

Methyl gallate, the active component of *Sanguisorba officinalis* L., has a therapeutic effect on ulcerative colitis, and its mechanism may be related to regulating polarization of macrophages, inhibiting activation of the TLR4/NF-κB signaling pathway and apoptosis of colon cells. The advantages of this study lie in clarifying the anti-inflammatory effect of *Sanguisorba officinalis* L. on the pharmacodynamic basis and mechanism and providing a new treatment option for UC with low cost and few side effects.

## Figures and Tables

**Figure 1 ijms-23-14024-f001:**
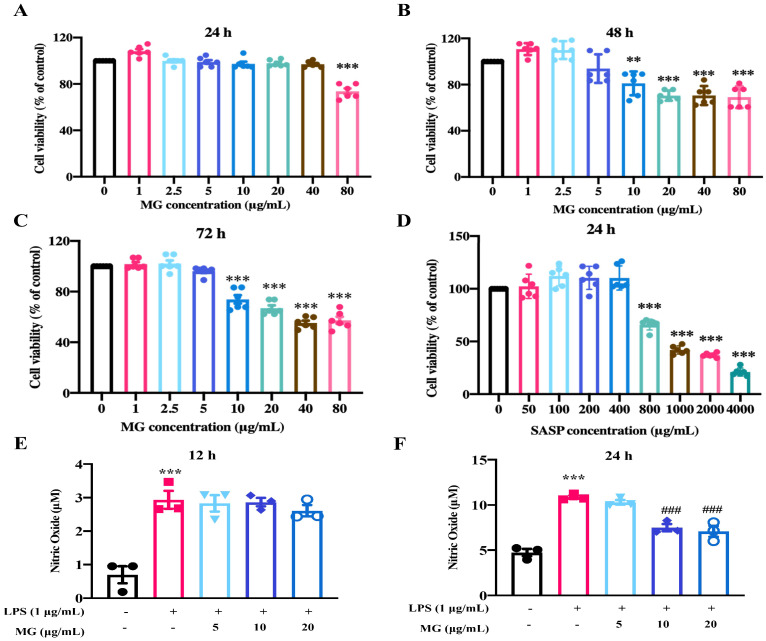
MG alleviates the inflammatory response of LPS-stimulated RAW264.7 cells. (**A**–**D**) Effects of different concentrations of MG and SASP on the viability of RAW264.7 cells. (**E**,**F**) Effect of NO production in RAW264.7 cells co-stimulated by MG and LPS for 12 h and 24 h. Data were presented as the means ± SEMs of three independent experiments. ** *p* < 0.01, and *** *p* < 0.001 vs. the normal group; ### *p* < 0.001 vs. the group treated with only LPS.

**Figure 2 ijms-23-14024-f002:**
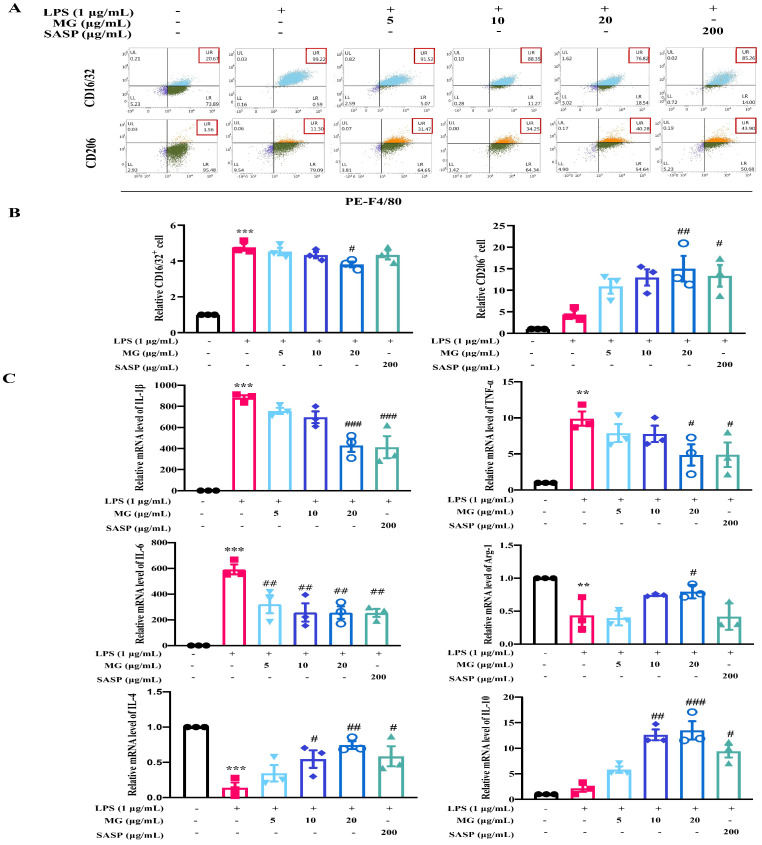
Effect of MG on RAW264.7 cell polarization. (**A**,**B**) Flow cytometry analysis of MG on RAW264.7 cell polarization in vitro. (**C**) qPCR analysis of MG effect on polarization-related mRNA levels in RAW264.7 cells. Data were presented as the means ± SEMs of three independent experiments. ** *p* < 0.01, and *** *p* < 0.001 vs. the normal group; # *p* < 0.05, ## *p* < 0.01, and ### *p* < 0.001 vs. the group treated with only LPS.

**Figure 3 ijms-23-14024-f003:**
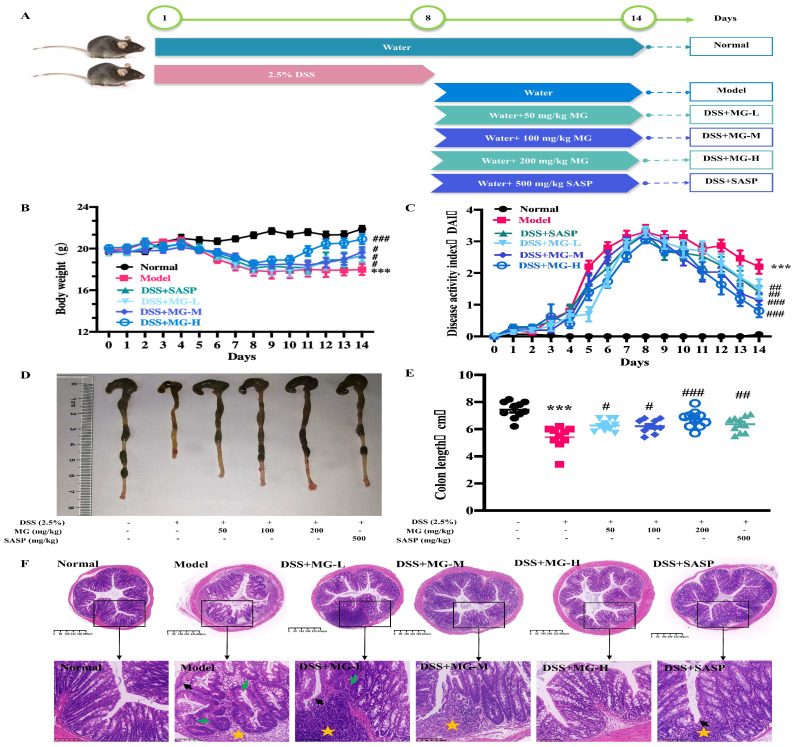
Effect of MG against UC caused by DSS. (**A**) Modeling and drug administration scheme of MG in DSS-induced mice. (**B**) The changes in body weight from each group of mice were measured. (**C**) Disease activity index (DAI) during the disease process. (**D**,**E**) Colon length of representative mice among different treatment groups. (**F**) Representative sections of hematoxylin and eosin (H&E) staining images of colon tissue (40×, scale bar: 400 μm) (200×, scale bar: 100 μm). Asterisks indicate interstitial lymphocytes infiltrated and clustered, black arrows indicate basal layer thickening and villous defects, green arrows indicate shortening, branching, and coiling of crypt structures. *** *p* < 0.001 vs. the normal group; # *p* < 0.05, ## *p* < 0.01, and ### *p* < 0.001 vs. the group treated with the DSS group.

**Figure 4 ijms-23-14024-f004:**
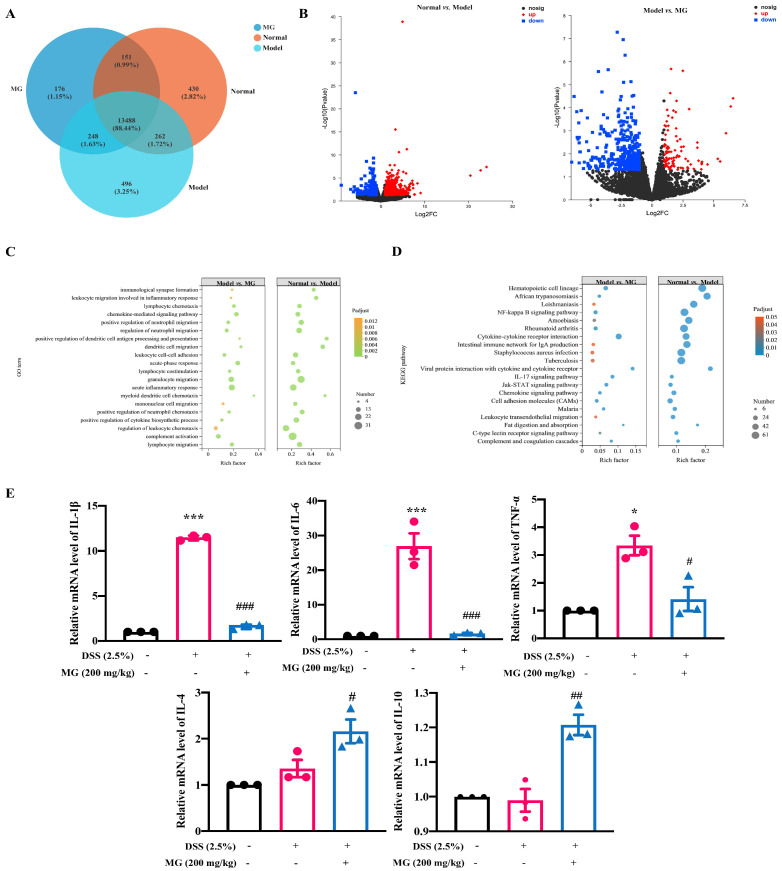
Transcriptome and qPCR analyses indicate the underlying biological function of MG in DSS-induced UC. (**A**) Venn diagram of the three groups. (**B**) Volcano diagram of the difference in gene expression. (**C**,**D**) GO and KEGG multi-gene enrichment analysis of differentially expressed genes. (**E**) Data were presented as the means ± SEMs of three independent experiments. * *p* < 0.05 and *** *p* < 0.001 vs. the normal group; # *p* < 0.05, ## *p* < 0.01, and ### *p* < 0.001 vs. the DSS group.

**Figure 5 ijms-23-14024-f005:**
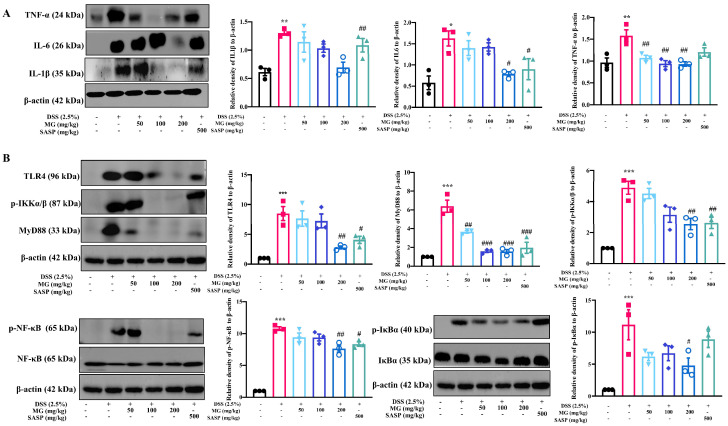
MG inhibites the expression of inflammatory cytokines (**A**) and TLR4/NF-κB signaling pathway activation (**B**) in vivo. Data were presented as the means ± SEMs of three independent experiments. * *p* < 0.05, ** *p* < 0.01, and *** *p* < 0.001 vs. the normal group; # *p* < 0.05, ## *p* < 0.01 vs. the DSS group.

**Figure 6 ijms-23-14024-f006:**
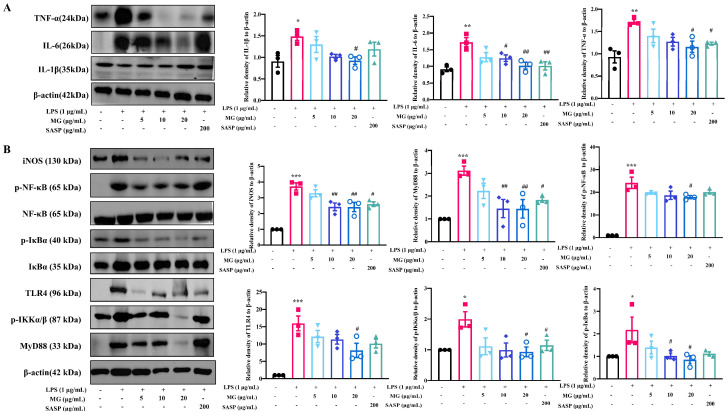
MG inhibites the expression of inflammatory cytokines (**A**) and TLR4/NF-κB signaling pathway activation (**B**) in vitro. Data were presented as the means ± SEMs of three independent experiments. * *p* < 0.05, ** *p* < 0.01, and *** *p* < 0.001 vs. the normal group; # *p* < 0.05, ## *p* < 0.01, and ###*p* < 0.001 vs. the group treated with only LPS.

**Figure 7 ijms-23-14024-f007:**
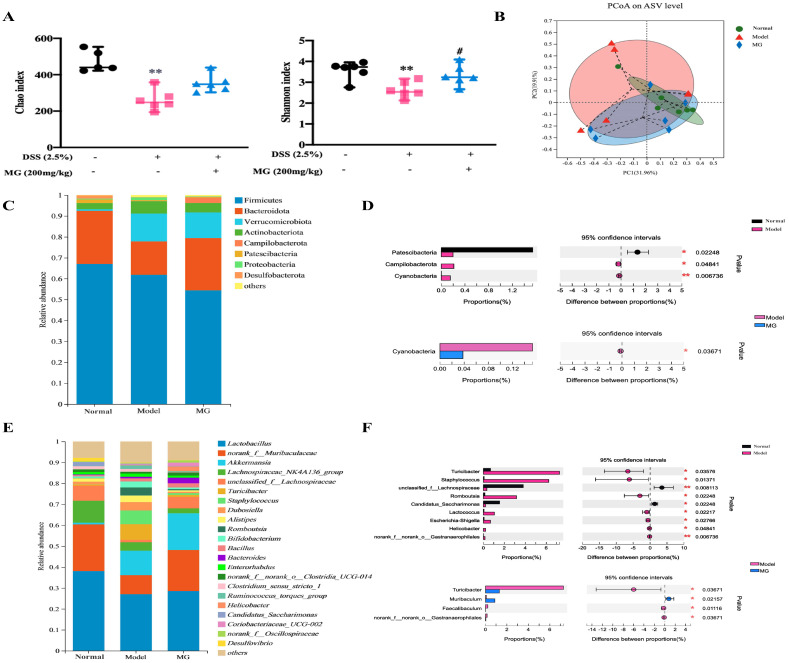
16S rRNA sequencing for the impact of MG on gut microbiota detection. (**A**) Alpha diversity indexes of each group (*n* = 6, ** *p* < 0.01 vs. the normal group; # *p* < 0.05 vs. the DSS group). (**B**) β-diversity evaluated using weighted UniFrac-based PCoA. (**C**,**D**) Relative abundances of microbiota constituents at the phylum level. * *p* < 0.05, ** *p* < 0.01. (**E**,**F**) Relative abundances of microbiota constituents at the genus level. * *p* < 0.05, ** *p* < 0.01.

**Figure 8 ijms-23-14024-f008:**
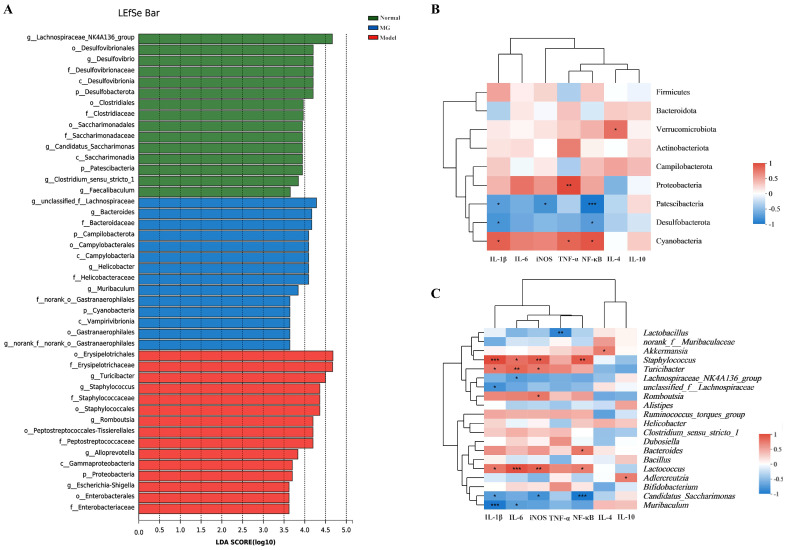
Correlation analysis between gut immunity inflammatory cytokines and gut microbiota. (**A**) Differentially enriched gut microbiota in three groups of mice by linear discriminant analysis (LDA) at the genus level. (**B**) Correlation matrix showing the strength of the correlation between gut immunity inflammatory cytokines and gut microbiota in the colon at the phylum level. (**C**) Correlation between gut immunity inflammatory cytokines and gut microbiota in the colon at the genus level. Spearman r values range from −1 (blue) to 1 (red). * *p* < 0.05, ** *p* < 0.01, and *** *p* < 0.001 vs. the normal group.

**Figure 9 ijms-23-14024-f009:**
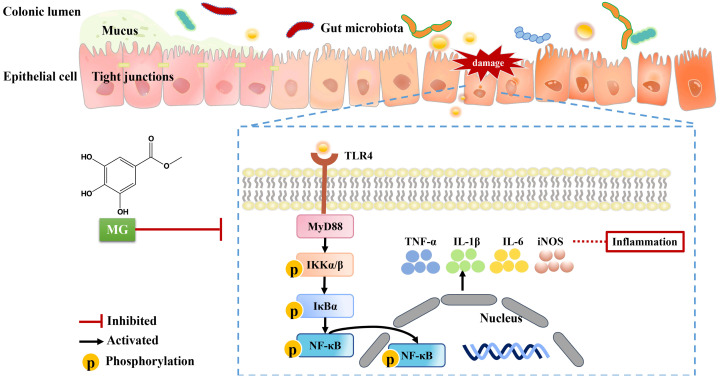
Action mode and potential mechanism of MG in treating UC.

## Data Availability

All figures and data used to support this study are included within this article.

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
