# Peer review of "Methyl Gallate Alleviates Acute Ulcerative Colitis by Modulating Gut Microbiota and Inhibiting TLR4/NF-κB Pathway"

_ijms, 2022, doi:10.3390/ijms232214024_

Round 1

Reviewer 1 Report

Please see my comments in the enclosed file.

Author Response

Dear Reviewer Expert:

 Thank you very much for the prompt review process and excellent comments. We greatly appreciate the time and efforts that you have spent on it. We are submitting the revised manuscript entitled “Methyl Gallate Alleviates Acute Ulcerative Colitis by Modu-lating Gut Microbiota and Inhibiting TLR4/NF-κB Pathway” (ID: ijms-1971505).

We have carefully considered your excellent comments and suggestions, and addressed each of the concerns in response to the comments (see point by point response). We have revised the manuscripts based on your comments and carefully checked throughout the manuscript and corrected the language errors. Our point-by-point responses to the comments (in blue) are shown below (in red).

  1. Colon isolated from DSS-treated mouse (Fig. 3D) does not exhibit any apparent signs of inflammation other than shortening and diarrhea. Could you please provide more photos of DSS-treated colons taken in better lighting for more detailed assessment?

Response: Thanks for your scientific comment. We collected phenotypic data according to the evaluation criteria of DSS-induced UC in the reference, the exact inflammation and lesions were displayed by analyzing pathological sections of colon. In addition, we will provide pictures of hematochezia in mice after DSS intervention (Supplementary Figure 2A), as well as pictures of all mice's colons (Supplementary Figure 3A).

  1. Fig 3F, microscopic photo: authors claim that the fibrosis is marked with asterisks. Those are submucosal sites infiltrated by immune cells rather than sites of fibrosis. This is apparent in the DSS+MG-L photo where tissue is cut through Peyer’s patch resulting in strong staining of accumulated immune cells. This needs to be corrected.

Response: Thanks for your careful review. It is a great suggestion that we have checked the marks of organizational positions, we searched the information [1] and agree with you that the asterisks-marked positions are submucosal sites infiltrated by immune cells, we made some mistakes in processing the data. After our verification, we have made corresponding modifications (Page 6, line 236-237). In the normal group, the colon epithelium of mice is intact, there is no ulcer or hyperplasia on the mucosal surface, and the crypt structure is regular and clear, and there is no obvious inflammatory infiltration. In DSS-induced model group, there was obvious inflammation between basal layer and mucosal muscle, and a large number of interstitial lymphocytes infiltrated and clustered (marked by asterisk), resulting in basal layer thickening and villous defects (marked by black arrow), then, crypt structure come into shortening, branching and curling (marked by green arrow).

  1. Perše M. and Cerar A. Dextran sodium sulphate colitis mouse model: traps and tricks. J Biomed Biotechnol. 2012, 718617. doi:10.1155/2012/718617
  2. What is the detailed interpretation of data presented on fig 4(A-D)? What conclusions do the authors infer from it? For example, what is the biological meaning of the fact that 248 (1.63%) genes are shared between MG and model group vs. 262 (1.72%) for model and normal groups?

Response: Thanks for your scientific comment. We are very sorry that there is no detailed interpretation of data presented on Fig 4(A-D), in the revised manuscript, we would interpret the results of Fig 4(A-D) in detail (Page 6, line 258-274. Page 12, line 463-472).

Fig 4A is Venn diagram between sample groups, which can obtain co-expressed and differentially expressed genes or transcripts among groups, and the value represents the number of common and differentially expressed genes or transcripts among different groups. Quantifying the proportion of expressed genes makes it more intuitive to compare the differences of expression among groups, which indicates that our subsequent analysis is based on all expressed genes or transcripts data obtained by this transcription sequencing.

In Fig 4B, negative binomial distribution software such as DESeq2, DEGseq and edgeR were used, and P<0.05 and |log2FC|≥1 were used for the screening conditions. It can be seen that the differentially expressed genes is symmetrically distributed on the volcano diagram, and the closer the point scattered to the edge, the more significant the differential genes expressed. According to the statistical analysis results of differentially expressed genes. Comparing the gene expression of the control group and DSS-induced model mice, a total of 1159 differentially expressed genes were screened, among which 814 genes were significantly up-regulated and 315 genes were significantly down-regulated. Compared with DSS model mice, 522 genes in the colon of the drug MG group were significantly changed, among which 100 genes were significantly up-regulated and 422 genes were significantly down-regulated. We further screened out the genes with significant expression for subsequent study.

Fig 4C-D shows the functional enrichment analysis of the obtained differentially expressed genes. We further clarified the functions of differentially expressed genes through the enrichment analysis of GO and KEGG pathways. No matter in the model group or the drug group, the differentially expressed genes are significantly enriched in the biological processes such as granulocyte migration, the regulation of acute inflammatory reaction, the regulation of dendritic cell antigen processing, presentation and migration, and the regulation of neutrophil migration. Similarly, multigene enrichment analysis of KEGG pathway showed significant enrichment in NF-κB signaling pathway, intestinal immune network producing immunoglobulin A (IgA), staphylococcus aureus infection, complement and coagulation cascade, hematopoietic cell lineage, and IL-17 signaling pathway. The above enrichment results suggest that treatments with DSS and MG affect the ex-pression of pathways linked to the regulation of inflammatory reactions and NF-κB signaling, which may be related to the potential molecular mechanism of MG in anti-colitis.

  1. Fig 4(C-D): Please explain the data presented on those figures. What do you mean by “Model vs. MG” and “Normal vs. Model”? Does it mean that certain pathways are enriched/more active in former vs the latter or vice versa?

Response: Thanks for your careful comment. "Model vs. MG” and “Normal vs. Model" means that the two groups are compared with each other, this cannot reflect the enriched/more active relationship between them. The figure mainly focuses on the function enrichment of the obtained differentially expressed genes. Through the enrichment analysis of the GO and KEGG pathways of differentially expressed genes, the possible mechanisms involved is further to discuss.

  1. Authors state that “The differentially expressed genes were significantly enriched in biological processes, including granulocyte migration, regulation of acute inflammatory response, and regulation of neutrophil migration in both the model and treatment groups”. How do you interpret the fact that the same processes are enriched/more active in both model and MG groups? Are those groups similar in terms of activity of these processes?

Response: Thanks for your scientific comment. As we mentioned in the manuscript, "The differentially expressed genes were significantly enriched in biological processes, including granulocyte migration, Regulation of acute inflammatory response, and regulation of neutral migration in both the model and treatment groups ", which was deduced from Fig 4(C-D). This data does not show that the model group and MG group have the same or similar trend in the mentioned biological processes, but the genes related to the function of this pathway are significantly different in both groups. In the following, qPCR and WB were conducted to verify the specific regulatory expression of key genes in inflammatory response and other pathways.

  1. Authors state that “The PCoA analysis revealed that MG could increase the gut microbiota in DSS-induced mice and re-establish them to a normal microbial community (Figure 7B).” Could you please elaborate on how did you infer this from the PCoA presented on fig 7B?

Response: Thanks for your scientific comment. After checking, we agree with your idea, and we are sorry for making mistakes in analyzing the data. We have added the details and made corresponding modification in revised manuscript (Page 9, line 387-394). As we can see in Figure 7B, the principal coordinate analysis (PCoA) based on Bray Curtis distance was used for visualized analysis of the difference in intestinal microbiota composition. Compared with the control group, the distribution of PCoA along PC2 among samples changed obviously in DSS-induced model group, indicating that the differences were obvious in intestinal microbiota composition, however, MG treatment could reduce the difference of microbiota composition, so we could infer that MG could increase the diversity and richness of intestinal microbiota and improve the difference of microbiota structure in mice with UC.

However, after we checked it out, the results that can truly show "MG could increase the gut microbiota in DSS-induced mice and re-establish them to a normal microbial community" should correspond to Figure 7 C-F, the results of Figure 7 C-F analyze the differences of intestinal flora at the level of phylum and genus respectively, which could better confirm the analysis results of this paragraph, so we have made corresponding adjustments (Page 10, line 415-420).

Sincerely,

Prof. Jianming Wu, Ph.D

Education Ministry Key Laboratory of Medical Electrophysiology

Sichuan Key Medical Laboratory of New Drug Discovery and Drugability Evaluation

Southwest Medical University

Reviewer 2 Report

In the manuscript entitled "Methyl Gallate Alleviates Acute Ulcerative Colitis by Modulating Gut Microbiota and Inhibiting TLR4/NF-κB Pathway", the alleviative effects of MG on UC were investigated in vitro and in vivo. The presented data indicated the oral administration of MG as a potential therapeutic strategy for UC. To explore the mechanisms involved, the TLR4/NF-κB signaling was detected in vitro and in vivo, and the transcriptome, 16S rRNA was analyzed. This work gives us a glimpse of this topic, however, there are several concerns to be addressed.

- The 16S RNA, transcriptome data can provide some clues. It is suggested to probe a certain mechanism deeply, rather than detect many aspects, but the data need to be strengthened.

- The signaling molecules expression and activation was analyzed by western blotting, without any rescue study. It is not solid enough to support the conclusions.

- The western data in vitro seems very consistent with that in vivo. There is no microbial influence on in vitro experiments. How to explain the conclusion “MG rectified gut microbiota imbalance in ulcerative colitis”?

- The resolution of Fig. 2, 4, 7, 8 are too low.

- The detail of 16S, transcriptome should be provided. The expression data should be submitted to database and made publicly available. And the different expression data should be provided in this manuscript at least.

- Some figures seem tricky. Fig. 2A, different gating was seen in the same protein. Fig.5 and 6, the photos are not consistent with those in the original images. Fig. 6b, its unbelievable to see the p-/total IkB in one gel with different molecular weights.

Author Response

Dear Reviewer Expert:

 Thank you very much for the prompt review process and excellent comments. We greatly appreciate the time and efforts that you have spent on it. We are submitting the revised manuscript entitled “Methyl Gallate Alleviates Acute Ulcerative Colitis by Modu-lating Gut Microbiota and Inhibiting TLR4/NF-κB Pathway” (ID: ijms-1971505).

We have carefully considered your excellent comments and suggestions, and addressed each of the concerns in response to the comments (see point by point response). We have revised the manuscripts based on your comments and carefully checked throughout the manuscript and corrected the language errors. Our point-by-point responses to the comments (in blue) are shown below (in red).

  1. The 16S RNA, transcriptome data can provide some clues. It is suggested to probe a certain mechanism deeply, rather than detect many aspects, but the data need to be strengthened.

Response: Thanks for your scientific comment. We have added the details in revised manuscript (Page 6, line 258-273. Page 9, line 375-381. Page 12, line 463-485). By analyzing the results of 16S rRNA detection of colon contents, it can be known that at the level of phylum, the abundance of Patescibacteria, Campilobacterota and Cyanobacteria are significantly different between the control group and the model group (P<0.05). The abundance of Cyanobacteria can be significantly reduced after MG treatment. Some research results show that the abundance of Cyanobacteria is obviously increased in premature senility [1], chronic asthma [2], septic shock [3], diarrhea [4] and immune diseases [5], which indicates that MG has treating effects on diarrhea and immune diseases. At the genus level, we found that Lactobacillus, norank_f__Muribaculaceae and unclassified_f__Lachnospiraceae were the main microbial groups in the feces of healthy mice, the model group was rich in Turicibacter and Erysipelotrichales, Staphylococcaceae, while MG-treated group enriched unclassified_f__Lachnospiraceae and Muribaculum, and increased the ability to produce beneficial metabolites [6]. Therefore, it is demonstrated that MG could improve the intestinal microbiota related to UC and it could produce beneficial metabolites, which could support the treatment group of MG for UC.

At the same time, transcriptome sequencing was used to analyze the obtained differentially expressed genes. We further clarified the functions of differentially expressed genes through the enrichment analysis of GO and KEGG pathways. No matter in the model group or the drug group, the differentially expressed genes are significantly enriched in the biological processes such as granulocyte migration, the regulation of acute inflammatory reaction, the regulation of dendritic cell antigen processing, presentation and migration, and the regulation of neutrophil migration. Similarly, multigene enrichment analysis of KEGG pathway showed significant enrichment in NF-κB signaling pathway, intestinal immune network producing immunoglobulin A (IgA), staphylococcus aureus infection, complement and coagulation cascade, hematopoietic cell lineage, and IL-17 signaling pathway. The above enrichment results suggest that treatments with DSS and MG affect the ex-pression of pathways linked to the regulation of inflammatory reactions and NF-κB signaling [7-9], which may be related to the potential molecular mechanism of MG in anti-colitis.

Reference:

  1. Merga Y., Campbell B.J., Rhodes J.M. Mucosal barrier, bacteria and inflammatory bowel disease: possibilities for therapy. Dig Dis. 2014, 32, 475-83. doi:10.1159/000358156
  2. Cleynen I., Laukens D. Cellular diversity in the colon: another brick in the wall. Nat Rev Gastroenterol Hepatol. 2019, 16, 391-392. doi:10.1038/s41575-019-0161-7
  3. Martini E., Krug S.M., Siegmund B., Neurath M.F., Becker C. Mend Your Fences: The Epithelial Barrier and its Relationship with Mucosal Immunity in Inflammatory Bowel Disease. Cell Mol Gastroenterol Hepatol. 2017, 4(1), 33-46. doi:10.1016/j.jcmgh.2017.03.007
  4. Coskun M. Intestinal epithelium in inflammatory bowel disease. Front Med (Lausanne). 2014, 1, 24. doi:10.3389/fmed.2014.00024
  5. Kotla N.G., Isa I.L.M., Rasala S., Demir S., Singh R., Baby B.V., Swamy S.K., Dockery P., Jala V.R., Rochev Y., Pandit A. Modulation of Gut Barrier Functions in Ulcerative Colitis by Hyaluronic Acid System. Adv Sci. 2022, 9(4), e2103189. doi:10.1002/advs.202103189
  6. Tavakoli P., Vollmer-Conna U., Hadzi-Pavlovic D., Grimm M.C. A Review of Inflammatory Bowel Disease: A Model of Microbial, Immune and Neuropsychological Integration. Public Health Rev. 2021, 42, 1603990. doi:10.3389/phrs.2021.1603990
  7. Xiong T., Zheng X., Zhang K., Wu H., Dong Y., Zhou F., Cheng B., Li L., Xu W., Su J., Huang J., Jiang Z., Li B., Zhang B., Lv G., Chen S. Ganluyin ameliorates DSS-induced ulcerative colitis by inhibiting the enteric-origin LPS/TLR4/NF-κB pathway. J Ethnopharmacol. 2022, 289, 115001. doi:10.1016/j.jep.2022.115001
  8. Feng Z., Zhou P., Wu X., Zhang J., Zhang M. Hydroxysafflor yellow A protects against ulcerative colitis via suppressing TLR4/NF-κB signaling pathway. Chem Biol Drug Des. 2022, 99(6), 897-907. doi:10.1111/cbdd.14045
  9. Wang Y., Tang Q., Duan P., Yang L. Curcumin as a therapeutic agent for blocking NF-κB activation in ulcerative colitis. Immunopharm Immunot. 2018, 40(6), 476-482. doi:10.1080/08923973.2018.1469145
  10.  

2. The signaling molecules expression and activation was analyzed by western blotting, without any rescue study. It is not solid enough to support the conclusions.

Response: Thanks for your careful comment. We are sorry that we did not do rescue study and we would like to further explain our model. Primarily, LPS, the modeling agent we used, could be recognized by Toll-like receptors and induce inflammatory response [1]. Under the stimulation of LPS, macrophages can express CD14 and TLR4 molecules, and the induced signals are further transmitted downward through MyD88 dependent and independent signaling pathways, and the downstream NF-κB is activated, thus activating inflammatory signaling pathways and promoting the release of proinflammatory cytokines. RAW264.7, as one of macrophages, is divided into M1 type and M2 type. M1 type mainly plays a proinflammatory role, which is usually activated by Toll-like receptors and often stimulated by interferon-γ, LPS and TNF-α. M2 type plays an anti-inflammatory role in immune response, it can inhibit inflammation and promote the recovery of damaged tissues [2-4].

In this experiment, we used LPS, a TLR4 inducer, as a modeling agent to differentiate macrophage RAW264.7 into M1 type, so as to construct an inflammatory model in vitro. In our experiment in vitro, we have verified that the cell model constructed by LPS was successful (Figure 1 A-F). After that, we screened MG from the monomer compound library, which can induce M1 cells constructed by LPS to polarize into M2 cells, thus playing an anti-inflammatory role. We proved this transformation process by flow cytometry in vitro, and the expression of inflammatory factors was verified to be decreased by Griess reagent kit (Figure 2 A-C). At the same time, Western Blot showed that the expression of TLR4 and downstream inflammatory signal pathway in LPS-induced model group was significantly enhanced. However, the expression of those in MG-treated group decreased, which indicated that MG could not only inhibit the expression of inflammatory factors induced by LPS, but also inhibit the activation of TLR4 induced by LPS (Figure 6). In addition, some studies have shown that the expression of TLR4 in colon tissues of mice with enteritis is obviously increased [5,6], and the severity of UC model mice after TLR4 gene knockout is significantly weakened after modeling [7]. WB experiment in vivo also proves that MG could inhibit TLR4 signal pathway (Figure 5).

Among many studies on UC in vivo and in vitro, LPS, the inducer of TLR4, is a classic inflammatory modeling agent in vitro, and DSS is also a classic model for UC mice in vivo. We used the two classic models to intervene, and the success of the model construction was verified. Phenotypes in vivo and in vitro indicated the therapeutic effect of MG, and MG also inhibited the expression of TLR4 and inflammatory pathway. We think that if the over-expression of inflammatory pathway is used for rescue research, the cells would suffer from overexpression of inflammation, and the model group would not survive and lack of comparison. Therefore, we chose to add inflammatory pathway inhibitor TAK242 for rescue research (Page 8-9, line 359-370). We conducted experiments in five groups in vitro, namely the control group, LPS-induced model group, MG-treated group, inhibitor-treated group, drug and inhibitor co-intervention group. As we can be seen in Supplementary Figure 1C, The expression of TLR4, downstream pathway p-IκBα and p-NF-κB in MG-treated group and inhibitor-treated group have no significant difference, but they are lower than those in model group, indicating that MG has the same effect of anti-inflammatory pathway expression with TLR4-inhibitor, and the expression of inflammatory signal pathway in co-intervention group is similar with that in MG group, indicating that inhibitor could competitively inhibit TLR4 target with MG and produce the same effect, which reflects that MG does exert its effect through TLR4, thereby inhibiting the expression of downstream inflammatory signaling pathway and inflammatory factors.

Combined with several references, the establishment of the classic UC model [8-10] and the data we added, we think that the difference between model establishment and administration both before and after can prove that the active component MG of Sanguisorba officinalis L. has therapeutic effect on UC, and its mechanism involves TLR4/NF-κB.

  1. Triantafilou M. and Triantafilou K. Lipopolysaccharide recognition: CD14, TLRs and the LPS-activation cluster. Trends Immunol. 2002, 23, 301-4. doi:10.1016/s1471-4906(02)02233-0
  2. Wynn T.A., Chawla A. and Pollard J.W. Macrophage biology in development, homeostasis and disease. Nature. 2013, 496, 445-55. doi:10.1038/nature12034
  3. Tarassishin L., Bauman A., Suh H.S., Lee Sunhee C. Anti-viral and anti-inflammatory mechanisms of the innate immune transcription factor interferon regulatory factor 3: relevance to human CNS diseases. J Neuroimmune Pharmacol. 2013, 8, 132-44. doi:10.1007/s11481-012-9360-5
  4. Li X. Gong J., Shi Y., Liu C., Peng Y. In vitro expression of CD14 protein and its gene in Kupffer cells induced by lipopolysaccharide. Hepatobiliary Pancreat Dis Int. 2003, 2(4), 571-5.
  5. Stahl M., Ries J., Vermeulen J., Yang H., Sham H., Crowley S.M., Badayeva Y., Turvey S.E., Gaynor E.C., Li X., Vallance B.A. A novel mouse model of Campylobacter jejuni gastroenteritis reveals key pro-inflammatory and tissue protective roles for Toll-like receptor signaling during infection. PLoS Pathog. 2014, 10(7), e1004264. doi:10.1371/journal.ppat.1004264
  6. Chen L., Lin M., Zhan L., Lv X. Analysis of TLR4 and TLR2 polymorphisms in inflammatory bowel disease in a Guangxi Zhuang population. World J Gastroenterol. 2012, 18(46), 6856-60. doi:10.3748/wjg.v18.i46.6856
  7. Eastaff-Leung N., Mabarrack N., Barbour A., Cummins A., Barry S. Foxp3+ regulatory T cells, Th17 effector cells, and cytokine environment in inflammatory bowel disease. J Clin Immunol. 2010, 30, 80-9. doi:10.1007/s10875-009-9345-1
  8. Liu S., Cao Y., Ma L., Sun J., Ramos-Mucci L., Ma Y., Yang X., Zhu Z., Zhang J., Xiao B. Oral antimicrobial peptide-EGCG nanomedicines for synergistic treatment of ulcerative colitis. J Control Release. 2022, 347, 544-560. doi:10.1016/j.jconrel.2022.05.025
  9. Gao C., Zhou Y., Chen Z., Li H., Xiao Y., Hao W., Zhu Y., Vong C., Farag M.A., Wang Y., Wang S). Turmeric-derived nanovesicles as novel nanobiologics for targeted therapy of ulcerative colitis. Theranostics. 2022, 12, 5596-5614. doi:10.7150/thno.73650
  10. Liu C., Yan X., Zhang Y., Yang M., Ma Y., Zhang Y., Xu Q., Tu K., Zhang M. Oral administration of turmeric-derived exosome-like nanovesicles with anti-inflammatory and pro-resolving bioactions for murine colitis therapy. J Nanobiotechnology. 2022, 20(1), 206. doi:10.1186/s12951-022-01421-w 3.The western data in vitro seems very consistent with that in vivo. There is no microbial influence on in vitro experiments. How to explain the conclusion “MG rectified gut microbiota imbalance in ulcerative colitis”?

Response: Thanks for your careful comment. We are sorry that we did not consider the effects of microorganisms in vitro when designing the experiment. Here, we’d like to provide more information to support our argument "MG rectified gut microbiota imbalance in ulcerative colitis", and we also made some supplements and explanations in the revised manuscript (Page 2, line 122-131. Page 12, line 486-494).

Primarily, there are some studies confirmed that a series of gut microbiota have protective or invasive functions in UC, such as the protective effect of the Firmicutes that metabolize to produce butyrate [1] and the inflammatory effect of adhesion invading enterobacteria [2]. It is known that ulcerative colitis is related to the steady balance between host mucosal immunity and gut microbiota. Therefore, we examined the differences in the composition of the gut microbiota at different taxonomic levels by sequencing 16S rRNA gene amplicons of intestinal contents in vivo. Spearman rank correlation method was used to analyze the correlation hotspot map at phylum and genus levels, respectively, and we found that MG could regulate the disordered microbiotic structure of UC mice by promoting the increase of beneficial microorganisms and the decrease of some opportunistic pathogens, which is closely related to the expression of inflammatory factors in the intestine. It is our mistake that we did not further test these intestinal florae in vitro. In view of the influence of a specific gut microbiota was large and extensive on the inflammatory reaction in vitro, and the in-depth research of intestinal microorganisms also involves the changes of metabolites and pharmacokinetics [3-5], so we did not conduct in-depth research on a specific microorganism for the time being. Also, we focused more to the therapeutic effect of MG, a new drug, on UC. According to the experimental design of experimental references related to ulcerative colitis [6,7], we think that the microbial detection of intestinal contents in vivo could support this argument.

In addition, the methods and systems we used for detection and analysis are in accord with the classic system of DSS-induced UC model mice [8,9]. Secondly, our DSS-induced UC model mice conform to animal ethics, and the model is stable, which could be used to effectively observe the efficacy and safety of drugs in vivo. The establishment of a complete and mature intestinal model of UC is complicated and unstable in vitro, which is rarely reported at present. And it is because of the complexity and variability of the internal environment, the data detected is more representative in vivo. At the same time, in many related reports, the analysis of gut microbiota in UC model mice has become a key means to prove its recovery [10-12], so it is believed that the detection of gut microbiota had certain reliability.

  1. Machiels K., Joossens M., Sabino J., De P.V., Arijs I., Eeckhaut V., Ballet V., Claes K., Van I.F., Verbeke K., Ferrante M., Verhaegen J., Rutgeerts P., Vermeire S. A decrease of the butyrate-producing species Roseburia hominis and Faecalibacterium prausnitzii defines dysbiosis in patients with ulcerative colitis. Gut. 2014, 63, 1275-83. doi:10.1136/gutjnl-2013-304833
  2. Maldonado-Arriaga B., Sandoval-Jiménez S., Rodríguez-Silverio J., Lizeth Alcaráz-Estrada S., Cortés-Espinosa T., Pérez-Cabeza d.V.R., Licona-Cassani C., Gámez-Valdez J.S., Shaw J., Mondragón-Terán P., Hernández-Cortez C., Suárez-Cuenca J.A., Castro-Escarpulli G. Gut dysbiosis and clinical phases of pancolitis in patients with ulcerative colitis. Microbiologyopen. 2021, 10, e1181. doi:10.1002/mbo3.1181
  3. Li Z., Ma S., Wang X., Wang Y., Yan R., Wang J., Xu Z., Wang S., Feng Y., Wang J., Mei Q., Yang P., Liu L. Pharmacokinetic and gut microbiota analyses revealed the effect of Lactobacillus acidophilus on the metabolism of Olsalazine in ulcerative colitis rats. Eur J Pharm Sci. 2022, 175, 106235. doi:10.1016/j.ejps.2022.106235
  4. Su L., Mao C., Wang X., Li L., Tong H., Mao J., Ji D., Lu T., Hao M., Huang Z., Fei C., Zhang K., Yan G. Schisandra chinensisThe Anti-colitis Effect of Polysaccharide Is Associated with the Regulation of the Composition and Metabolism of Gut Microbiota. Front Cell Infect Microbiol. 2020, 10, 519479. doi:10.3389/fcimb.2020.519479
  5. Jourova L., Anzenbacher P., Matuskova Z., Vecera R., Strojil J., Kolar M., Nobilis M., Hermanova P., Hudcovic T., Kozakova H., Kverka M., Anzenbacherova E. in vitroGut microbiota metabolizes nabumetone: Consequences for its bioavailability in the rodents with altered gut microbiome. Xenobiotica. 2019, 49, 1296-1302. doi:10.1080/00498254.2018.1558310
  6. Tang B., Zhu J., Zhang B., Wu F., Wang Y., Weng Q., Fang S., Zheng L., Yang Y., Qiu R., Chen M., Xu M., Zhao Zi., Ji J. Therapeutic Potential of Triptolide as an Anti-Inflammatory Agent in Dextran Sulfate Sodium-Induced Murine Experimental Colitis. Front Immunol. 2020, 11, 592084. doi:10.3389/fimmu.2020.592084
  7. Qu Y., Li X., Xu F., Zhao S., Wu X., Wang Y., Xie J. Kaempferol Alleviates Murine Experimental Colitis by Restoring Gut Microbiota and Inhibiting the LPS-TLR4-NF-κB Axis. Front Immunol. 2021, 12, 679897. doi:10.3389/fimmu.2021.679897
  8. Katsandegwaza B., Horsnell W., Smith K. Inflammatory Bowel Disease: A Review of Pre-Clinical Murine Models of Human Disease. Int J Mol Sci. 2022, 23(16), undefined. doi:10.3390/ijms23169344
  9. Li M., Yang L., Mu C., Sun Y., Gu Y., Chen D., Liu T., Cao H. Gut microbial metabolome in inflammatory bowel disease: From association to therapeutic perspectives. Comput Struct Biotechnol J. 2022, 20, 2402-2414. doi:10.1016/j.csbj.2022.03.038
  10. Ordás I.; Eckmann L.; Talamini M.; Baumgart D.C.; Sandborn W.J. Ulcerative colitis. Lancet. 2012, 380, 1606–1619. doi: 10.1016/S0140-6736(12)60150-0
  11. Feuerstein J.D., Moss A.C., Farraye F.A. Ulcerative Colitis. Mayo Clin Proc. 2019, 94, 1357-1373. doi:10.1016/j.mayocp.2019.01.018

Feuerstein J.D., Cheifetz A.S. Crohn Disease: Epidemiology, Diagnosis, and Management. Mayo Clin Proc. 2017, 92, 1088-1103. doi:10.1016/j.mayocp.2017.04.010

4. The resolution of Fig. 2, 4, 7, 8 are too low.

Response: Thanks for your careful comment. We regret that we only noticed the requirements of the magazine and did not consider whether the clarity of the picture is suitable for reading, now we have modified the resolution of the pictures in order to have a better and clearer reading experience.

  1. The detail of 16S, transcriptome should be provided. The expression data should be submitted to database and made publicly available. And the different expression data should be provided in this manuscript at least.

Response: Thanks for your careful comment. After Analyzing the different expression data, we sorted out and summarized the genes obtained from all samples after transcription sequencing, and the expression of genes among groups, then we and summarized them in the Metadata spreadsheet (Supplementary Files). These Metadata spreadsheets provide more complete data supporting for results of transcriptome sequencing. We are uploading the raw data of 16s RNA and transcriptome sequencing to the GEO database. But it is sorry to say that due to the large data (all the raw data occupy the capacity of 41GB), our uploading speed is limited. Currently, it is still in the process of uploading (Our personalized upload space is: uploads/[email protected]_GXOhW0oq), and we need to get the reply from GEO database after uploading. Once successful, we will reply to the editor and you by email to provide more reference for our raw data. Of course, all the raw data and supplementary files are worthy for further exploration, and we hope to provide some accessible data references for interested public readers.

  1. Some figures seem tricky. Fig. 2A, different gating was seen in the same protein. Fig.5 and 6, the photos are not consistent with those in the original images. Fig. 6b, it’s unbelievable to see the p-/total IkB in one gel with different molecular weights.

Response: Thanks for your careful comment. In view of the problem of setting the axes in flow cytometry, we have made corresponding modifications. Our procedure is carried out under the same voltage condition, but the form of presentation has changed when exporting the data, so the settings of coordinate axes are different for different groups, and we ignored this point when screening the data, and now all the data have been unified. As for WB, we are so sorry that we made some mistakes in collating the raw data, we carefully checked the images presented, and the representative images of IL-1β in this group in Fig 5A were confused with one of the duplicate pictures, which has been corrected. p-NF-κB displayed in Fig 5B/Fig 6B has been revised accordingly. In Fig 6B, the antibody against p-IκBα was obtained from CST (Lot: 2859) at 40 kDa, and the antibody against total IkBα was obtained from proteintech (Lot: 10268-1-AP) at 36 kDa. The two antibodies were incubated in the same band, so p-/total IkB appeared in the same gel.

Sincerely,

Prof. Jianming Wu, Ph.D

Education Ministry Key Laboratory of Medical Electrophysiology

Sichuan Key Medical Laboratory of New Drug Discovery and Drugability Evaluation

Southwest Medical University

Round 2

Reviewer 1 Report

Authors addressed all my concerns. I have no further comments.

Author Response

Thanks a lot for your comment.

Reviewer 2 Report

Serious concerns have not been addressed.

Author Response

Dear Reviewer Expert,

Thank you very much for the prompt review process and excellent comments. We greatly appreciate the time and efforts you spent on it. We are submitting the revised manuscript entitled “Methyl Gallate Alleviates Acute Ulcerative Colitis by Modu-lating Gut Microbiota and Inhibiting TLR4/NF-κB Pathway” (ID: ijms-1971505).

We have carefully considered your excellent comment and suggestion, and addressed the concerns in response to the comment. We have revised the manuscripts based on your comments and carefully checked throughout the manuscript and corrected the errors. Our response to the comments (in blue) are shown below (in red).

  1. Serious concerns have not been addressed.

Response: Thanks for your careful comment. We have carefully considered your comments and suggestions, and addressed the concerns in response to the comments.

First of all, we sent the original database link on October 26th and we will provide the link again now. Here is the raw data link for 16s RNA and transcriptome sequencing: https://www.ncbi.nlm.nih.gov/geo/query/acc.cgi?acc=GSE216435, because it remains in private status, the private access token is qpovgkoonfetbqx. The database numbered GSE216435 includes two original databases: GSE216433 (16s RNA) and GSE216434 (RNA-seq). We hope you can have a more comprehensive understanding of the results through the database, at the same time, we would also make further exploration in the following experiment.

Next, we would further clarify the problems of our figures and data. In view of the problem of gating in flow cytometry, we have made further modifications. When we conducted the experiments of flow cytometry, we kept the same voltage condition for three repeated experiments, but due to the change of cell state during the experiments (the polarization of RAW264.7 cells were unstable), we adjusted the gating in different batches of experiments (the related voltage did not change), which led to different presentation forms when we exported the data. Then, due to our negligence, we hoped to show more representative data when we sorted out the data, and did not notice that the settings of experimental gates were different in different batches. Now, we have unified to ensure that the data of the same batch of experiments are displayed (Figure 2A and B, Figures can be found in the attachment). At the same time, in the last modification, we uploaded the raw data of flow cytometry in format of fcs, and we hope you can have a further understanding of our flow cytometry through these.

Similarly, in view of the discrepancy between the raw data uploaded and Figures 5/6, we confirm that there is no problem with the updated ones after strict check and correction one by one, the representative images of IL-1β in this group in Fig 5A were confused with one of the duplicate pictures, which has been corrected, and p-NF-κB displayed in Fig 5B/Fig 6B has been revised accordingly. At the same time, some of the bands look different from the original bands, because we stretched the pictures in order to ensure the unified effect of presentation, this is shown in the figure below, which represents the IκBα of Figure 5B. In this process, we only changed the length and width of the bands, and at the time of data analysis, we used the original and unprocessed bands.

As for the problem that p-/total- IkBα appears in the same band, we will make further clarification. First of all, we added the related method of western blot in manuscript (Page16, line 717-721, , Figures can be found in the attachment), through reviewing the methodology and literature, we ensured that our method was feasible [1], that is, after the band was detected with an ECL system and exposed to classic autoradiography film for the first time, the phosphorylated protein on the membrane would be stripped with Stripping Buffer, and the related membrane was incubated with the total antibody overnight at 4℃, and the previous operation was repeated. Meanwhile, the gray value of the target protein and phosphorylated protein was compared with that of β-actin, GAPDH or total protein. Since our phosphorylated- and total- antibodies come from different companies, their molecular weights are different. The anti-p-IκBα antibody was obtained from CST (Lot: 2859) at 40 kDa, and the anti-IκBα antibody was obtained from Proteintech (Lot: 10268-1-AP) at 36 kDa, and the stripping buffer we used at that time was not so effective (probably because the stripping buffer came from different batches), resulting in the p-IκBα was not completely stripped off. Under this condition, we incubated total-IκBα again, resulting in phosphorylated and total two proteins in the same bands during visualization. Therefore, 36kDa of total-IκBα (not a non-specific binding protein) and 40kDa of p-IκBα were found on the bands, we found this problem when sorting out the data. However, although there is no precedent for this, we thought that such bands might have some reference, so we chose to keep this band. After our re-experimental verification, we replaced the bands and analyzed the data of the newly replaced band (Figure 6B, , Figures can be found in the attachment). It can be seen more intuitively that there is a difference in molecular weights between the two proteins in the triplicate raw data of WB, and the trends of expression among different proteins in different groups are also consistent with those before.

Finally, we double-checked all the figures and readjusted the clarity of the figures, hoping that you can have a deeper understanding of our experimental results through our updated data and figures, and we are trying to make further exploration for this. If there are any questions, we are looking forward to further communication with you.

  1. Kim B. Western Blot Techniques. Methods Mol Biol. 2017, 1606:133-139. doi: 10.1007/978-1-4939-6990-6_9. PMID: 28501998.

Sincerely,

Jianming Wu, Ph.D
